# Effects of Compression and Porosity Gradients on Two-Phase Behavior in Gas Diffusion Layer of Proton Exchange Membrane Fuel Cells

**DOI:** 10.3390/membranes13030303

**Published:** 2023-03-04

**Authors:** Hao Wang, Guogang Yang, Qiuwan Shen, Shian Li, Fengmin Su, Ziheng Jiang, Jiadong Liao, Guoling Zhang, Juncai Sun

**Affiliations:** 1Marine Engineering College, Dalian Maritime University, Dalian 116026, China; 2Laboratory of Transport Pollution Control and Monitoring Technology, Beijing 100084, China

**Keywords:** gas diffusion layer, compression, porosity gradient, two-phase behavior, lattice Boltzmann method

## Abstract

Water management within the gas diffusion layer (GDL) plays an important role in the performance of the proton exchange membrane fuel cell (PEMFC) and its reliability. The compression of the gas diffusion layer during fabrication and assembly has a significant impact on the mass transport, and the porosity gradient design of the gas diffusion layer is an essential way to improve water management. In this paper, the two-dimensional lattice Boltzmann method (LBM) is applied to investigate the two-phase behavior in gas diffusion layers with different porosity gradients under compression. Compression results in an increase in flow resistance below the ribs, prompting the appearance of the flow path of liquid water below the channel, and liquid water breaks through to the channel more quickly. GDLs with linear, multilayer, and inverted V-shaped porosity distributions with an overall porosity of 0.78 are generated to evaluate the effect of porosity gradients on the liquid water transport. The liquid water saturation values within the linear and multilayer GDLs are significantly reduced compared to that of the GDL with uniform porosity, but the liquid water within the inverted V-shaped GDL accumulates in the middle region and is more likely to cause flooding.

## 1. Introduction

A proton exchange membrane fuel cell (PEMFC) is an energy power device that converts chemical energy from hydrogen and oxygen directly into electricity. The PEMFC has received worldwide attention because of its high efficiency, zero emissions, and low noise [1]. In the cathode of PEMFC, oxygen flows through the gas channel and gas diffusion layer (GDL) and then reacts with protons in the catalytic layer and produces water [2]. Inappropriate water management will result in dehydration of the proton exchange membrane and flooding of the electrode, which will seriously affect the performance and lifetime of the fuel cell. [3] The two-phase behavior in the gas diffusion layer is critical in the water management of the PEMFC. Generally, the components of the fuel cell are assembled by assembly pressure to ensure their gas tightness. GDL under the rib is compressed and deformed due to the assembly pressure, but the structural deformation of the GDL under the channel is relatively slight. The inhomogeneous deformation of the GDL by compression not only results in the stress concentration phenomenon at the interface between the GDL and the rib but also significantly affects the water transport process in the GDL. The pore region of the GDL is employed for reactive gas diffusion and liquid water removal, so the design of the GDL highly influences the two-phase behavior within the GDL [4]. Additionally, factors affecting water transport within the GDL also include the structural parameters, rib-channel width ratio, wettability, and microporous layer (MPL) [5].

Some experiments have investigated the two-phase behavior within the GDL; the most popular methods are neutron imaging [6,7] and X-ray imaging [8,9,10]. Neutron imaging was employed by Kulkarni et al. [6] to study the distribution of liquid water within the GDL at different compression ratios. Poorer water removal efficiency due to increased compression was observed, and a strong dependence of compression on water accumulation and removal within the GDL was revealed. Neutron imaging was used by Cho et al. [7] to visualize the distribution of liquid water in the PEMFC with lung-inspired and serpentine flow fields. The liquid water transport behavior for three commercial GDL materials, Freudenberg H2315 I6, Toray TGP-H-060, and SGL 24BA, was analyzed by Xu et al. [9]. Higher water saturation values in compressed GDL samples than in uncompressed GDL samples were observed by Ince et al. [10], which may be due to the fact that compression promotes the flow and accumulation of liquid water in the in-plane direction.

Benefiting from advances in computational power, numerical simulation methods have been widely adopted to study water transport within the PEMFC, such as volume of fluid (VOF) [11,12], pore network method (PNM) [13,14], direct numerical simulation (DNS) [15] and lattice Boltzmann method (LBM). A three-dimensional VOF model was developed by Shi et al. [11] to investigate the effect of MPL cracking characteristics on the two-phase mass transport mechanism in GDL. The VOF model was applied by Anyanwu et al. [12] to investigate the effect of compression on liquid water transport within the GDL and found that the two-phase behavior was influenced by the GDL contact angle, compression ratio, inlet pressure, and capillary pressure. The PNM model was adopted by Carrere et al. [13] and Gholipour et al. [14] to investigate the influence of operating temperature, rib width, and GDL thickness on water transport. Due to the simplicity of the algorithm, the easy handling of complex boundary conditions, the convenience of solving pressure, and the high parallelism, the LBM is popular for investigating the water transport inside the GDL. Jeon [16] and Yang et al. [17] adopted LBM to study the flow of liquid water due to different rib-channel widths and found that the two-phase behavior under the rib was the origin of the difference in fuel cell performance. Scholars have focused on the different wettability of fiber [18] and non-uniform distribution of PTFE [19,20,21,22,23], and the wettability of GDL has been demonstrated to have a significant effect on water transport. Hao and Cheng [18] compared highly hydrophobic GDLs with GDLs with nearly neutral wettability and considered wettability to be an important parameter controlling the two-phase behavior within the GDL. Ira et al. [19] inserted a hydrophilic layer into the GDL to improve water management. They also evaluated the two-phase behavior of the GDL when a proportion of fibers within the GDL were hydrophilic and found that the presence of hydrophilic fibers below the channels facilitated the removal of water, while their presence below the ribs led to the accumulation of water within the GDL [20]. Additionally, other scholars have adopted LBM to study the effects of GDL thickness [24], MPL [25,26], perforation [27], and fiber orientation [28] on water transport.

Compression significantly affects the porosity and water–gas diffusion paths of the GDL and, thus, the mass transport processes in the GDL. The stress–strain distribution between the fibers and the binder inside the compressed GDL was simulated by Xiao et al. [29] in a combination of finite element and explicit kinetic methods. They also evaluated the porosity, gas diffusivity, liquid water permeability, thermal conductivity, resistivity, and mechanical strength of paper-type and felt-type GDLs at different compression ratios (CRs) using XCT, compression tests, and LBM [30]. The mechanical properties of GDL under cyclic loading were studied by Meng et al. [31], and a non-linear intrinsic model describing the compression properties was proposed. Macro-scale models have also been developed by some scholars to study the effect of GDL compression on the transport processes within PEMFC [32,33,34,35]. Zhang et al. [32] developed a comprehensive two-dimensional two-phase PEMFC model coupling solid mechanics, heat and mass transfer, and electrochemical reactions and investigated the effects of GDL deformation on coupled heat and mass transfer and liquid water distribution. The combined effect of the flow field and assembly pressure leads to the inhomogeneous distribution of porosity, mass concentration, and current density, and high assembly pressure results in more inhomogeneous oxygen and water concentrations [33]. In previous pore-scale studies, scholars have mainly focused on the effect of compression on the transport properties of GDL. The anisotropic transport properties of compressed Toray TGP-H-060 and Freudenberg H2315, such as diffusivity, electrical conductivity, and permeability, were investigated by Zhang et al. [36]. The effects of compression and anisotropy on the transport properties of reconstructed GDLs were investigated by Zhu et al. [37]. They found that the effective gas diffusion coefficient decreased with increasing compression ratio while the effective thermal conductivity and electrical conductivity increased. Lee et al. [38] revealed that the variation of fiber volume fraction with compression ratio has a large effect on the longitudinal elastic modulus. The effect of compression on tortuosity and permeability in the in-plane and through-plane directions of GDLs was investigated by Froning et al. [39].

In recent years, scholars have focused on designing the GDL pore distribution to improve water–gas transport within the GDL. Oh et al. [40] applied GDLs with different porosity to form a double-layered GDL substrate and found that the resulting pore gradient structure improved the PEMFC steady-state performance and enhanced the transient response. Macro-scale studies have also been carried out, and Huang et al. [41] found that the linear distribution of porosity increased the limiting current density and the oxygen usage in PEMFC with serpentine flow fields. Multilayer-type GDLs were designed by Kanchan et al. [42,43] along the in-plane and through-plane directions, and it was found that the lower porosity near the interface of the cathode catalyst layer resulted in higher current density and power density of the fuel cell. As the complex pore structure of GDLs has a significant impact on the mass transfer, developing mesoscale simulation methods that can capture the microstructure of GDLs is required for the structural design of GDLs. In our previous studies, we found that the porosity gradient distribution in the through-plane direction can increase the permeability of GDLs [44]. Habiballahi et al. [45] showed that when GDL porosity increases linearly from near the catalytic layer side to near the channel side, the average water saturation values of porosity gradient GDLs are lower than that of GDL with uniform porosity. A three-dimensional two-phase VOF model was developed by Shangguan et al. [46] to investigate water transport within V-shaped and inverted V-shaped porosity gradient GDLs. It was found that the higher the GDL porosity gradient, the fewer the water penetration paths and the smaller the water saturation and water flow rate within the porosity gradient GDL.

From the reported studies, it appears that GDL compression is an inevitable phenomenon in the assembly process of PEMFCs, and there is a lack of comprehensive studies on the two-phase transport and characteristics within compressed GDLs. There has been a trend to optimize mass transfer within GDLs by designing porosity gradients, while the two-phase behavior within porosity gradient GDLs has not been well studied. Further, for the designed porosity gradient GDLs, no studies have focused on their two-phase behavior under compression. Since both compression and porosity gradient distributions have influences on the flow path of water transport within a GDL, the location of water breakthrough into the channel, and the time for liquid water distribution to reach a steady state, the study of the two-phase behavior of porosity gradient GDLs under compression is particularly valuable in supporting PEMFC water management.

In the present paper, the pseudo-potential two-phase LB model proposed by Shan and Chen [47] is established to investigate the effects of compression and porosity gradients on the transport of liquid water. The developed two-phase LB model is validated with the bubble test and contact angle test. Different types of porosity gradients (linear, multilayer, and inverted V-shaped) are designed, and two gradients are considered for each type. The effects of the porosity gradient on the liquid water distribution and liquid water saturation within the GDL are evaluated at different compression ratios (0, 20%, 40%).

## 2. Numerical Approach

### 2.1. Pseudo-Potential Multi-Phase LB Model

The pseudo-potential LB model, proposed by Shan and Chen, assumes the existence of multiple fluids at each node and describes the behavior of each fluid separately with a distribution function. The separation between different phases is realized by the forces between the fluid components, and the wettability of the fluid on the solid surface is depicted by the forces between the fluid components and the solid wall. The evolving equations of the distribution function ($f_{i}^{k}$) and the equilibrium distribution function ($f_{i}^{k,eq}$) are expressed as follows [48]:(1)$$f_{i}^{k}(x+c_{i}\Delta t,t+\Delta t)=f_{i}^{k}(x,t)+\frac{\Delta t}{\tau^{k}}\left[f_{i}^{k,eq}(x,t)-f_{i}^{k}(x,t)\right]$$
(2)$$f_{i}^{k,eq}=w_{i}\rho^{k}\left[1+\frac{c_{i}\times u^{k,eq}}{c_{s}{}^{2}}+\frac{1}{2}\frac{{\left(c_{i}\times u^{k,eq}\right)}^{2}}{c_{s}{}^{4}}-\frac{1}{2}\frac{u^{k,eq}{}^{2}}{c_{s}{}^{2}}\right]$$
where Δ*t* denotes the time step, and *τ^k^* and *ρ^k^* represent the relaxation time and density of component *k*, respectively. *c_s_* is the lattice speed of sound. *w_i_* and *c_i_* are the weight factor and particle velocity vector, respectively. In this paper, the D2Q9 model is adopted and the corresponding *w_i_* and *c_i_* are given as follows:(3)$$w_{i}=\{\begin{array}{c} \begin{array}{c} 4/9\;\;i=1\\ 1/9\;\;i=2,3,4,5 \end{array}\\ 1/36\;i=6,7,8,9 \end{array}$$
(4)$$\left[\begin{array}{ccc} \begin{array}{ccc} \begin{array}{cccc} \begin{array}{cc} c_{1} & c_{2} \end{array} & c_{3} & c_{4} & c_{5} \end{array} & c_{6} & c_{7} \end{array} & c_{8} & c_{9} \end{array}\right]=\left[\begin{array}{c} \begin{array}{cccc} \begin{array}{cccc} \begin{array}{ccc} 0 & 1 & 0 \end{array} & -1 & 0 & 1 \end{array} & -1 & -1 & 1 \end{array}\\ \begin{array}{cccc} \begin{array}{cccc} \begin{array}{ccc} 0 & 0 & 1 \end{array} & 0 & -1 & 1 \end{array} & 1 & -1 & -1 \end{array} \end{array}\right]$$

The density and velocity (***u**^k^*) of each fluid can be obtained from their respective distribution functions. The equilibrium velocity (***u**^k,eq^*) in the equilibrium distribution function is redefined in the following way:(5)$$\rho^{k}=\sum_{i=1}^{9}f_{i}^{k}(x,t)$$
(6)$$\rho^{k}u^{k}=\sum_{i=1}^{9}f_{i}^{k}(x,t)c_{i}$$
(7)$$u^{k,eq}=u^{\prime }+\frac{\tau^{k}F^{k}}{\rho^{k}}$$
where ***u***′ is the combined velocity of the mixture and can be expressed as follows:(8)$$u^{\prime }=\frac{\sum_{k}\frac{1}{\tau^{k}}\sum_{i=1}^{9}f_{i}^{k}(x,t)c_{i}}{\sum_{k}\frac{1}{\tau^{k}}\rho^{k}}$$

***F****^k^* denotes total external forces of component *k*, including fluid–fluid interaction force ($F_{f-f}^{k}$) and fluid–solid interaction force ($F_{f-s}^{k}$):(9)$$F^{k}=F_{f-f}^{k}+F_{f-s}^{k}$$

The fluid–fluid interaction force of the component *k* at position ***x*** is defined as [49]:(10)$$F_{f-f}^{k}(x)=-\psi^{k}(\rho^{k}(x))\sum_{x^{\prime }}\sum_{\bar{k}}^{s}G^{k\bar{k}}(x,x^{\prime })\psi^{\bar{k}}(\rho^{\bar{k}}(x))(x^{\prime }-x)$$
where *ψ^k^* is the effective density of component *k* at position ***x*** and can be taken as $\psi^{k}(\rho^{k}(x))=\rho_{0}(1-\exp(-\rho^{k}/\rho_{0}))$. ***x****′* represents the surrounding nodes centered on position ***x***. $G^{k\bar{k}}$ is Green’s function at position ***x*** and when the interaction forces between adjacent and sub-adjacent lattice points are considered, $G^{k\bar{k}}$ can be expressed as:(11)$$G^{k\bar{k}}=\{\begin{array}{c} \begin{array}{c} 4g\left|x-x^{\prime }\right|=1\\ g\left|x-x^{\prime }\right|=\sqrt{2} \end{array}\\ 0\left|x-x^{\prime }\right|=0 \end{array}$$

Usually *g* takes a positive value and can be used to adjust the surface tension between different fluids. The fluid-solid interaction force can be expressed as:(12)$$F_{f-s}^{k}(x)=-\psi^{k}(\rho^{k}(x))\sum_{x^{\prime }}W(x,x^{\prime })s(x^{\prime })(x^{\prime }-x)$$
(13)$$W=\{\begin{array}{c} \begin{array}{c} 4w\left|x-x^{\prime }\right|=1\\ w\left|x-x^{\prime }\right|=\sqrt{2} \end{array}\\ 0\left|x-x^{\prime }\right|=0 \end{array}$$
where the values of *s*(***x***′) are 0 or 1 for fluid or solid at position ***x***′ respectively. *w* adjusts the interaction forces between fluid and solid. When *w* takes a positive value it means that the solid is a hydrophobic wall (contact angle greater than 90°) and a negative value for a hydrophilic wall (contact angle less than 90°). For the two-phase behavior discussed in this paper, the values of *w* taken for water and gas are opposite to each other.

### 2.2. Validation

The Young–Laplace law was used to verify the validity of the model established in this paper. A droplet of radius R is placed in the center of a 100 × 100 lu^2^ square computational domain and the rest of the domain is filled with gas. The initial density is set as follows: the droplet region is set to *ρ^ai^*^r^ = 1 × 10^−5^, *ρ^water^* = 2; the gas region is set to *ρ^water^* = 1 × 10^−5^, *ρ^air^* = 2 [50,51]. From the Young–Laplace law, it follows that the droplet will remain a steady circle under surface tension when the system reaches force equilibrium and that the pressure difference between the inside and outside of the droplet at a steady state satisfies the following relationship:(14)$$\Delta P=P_{g,in}-P_{g,out}=\frac{\sigma }{R}$$
(15)$$P=c_{s}{}^{2}\sum_{k}\rho^{k}+\frac{c_{s}{}^{2}}{2}\sum_{k,\bar{k}}G^{k\bar{k}}\psi^{k}\psi^{\bar{k}}$$

In the validation simulations, periodic boundary conditions are taken around the computational domain, and eight different cases of initial droplet radius are simulated in turn. Figure 1a shows the relationship between the pressure difference Δ*P* between the inside and outside of the droplet and the inverse of the droplet radius. The simulation results are fitted and found to be positively proportional, which shows that the numerical simulation results conform to the Young–Laplace law and validate the accuracy of the model.

Typically, the wettability exhibited macroscopically as the contact angle of the fluid on a solid wall; that is, the contact angle of the fluid on a hydrophilic wall is less than 90°, while greater than 90° on a hydrophobic wall. A semi-circular droplet with an initial radius of 15 lu is considered to be placed in the center of the bottom of a 100 × 100 lu^2^ computational domain, with solid boundaries on the top and bottom surfaces and periodic boundaries in the left and right planes. The shape of the droplet is constantly evolving due to the surface tension until it reaches a steady state. The variation of contact angle with w value is shown in Figure 1b, verifying that the model established in this paper can correctly deal with the wettability of a fluid on the solid surface.

### 2.3. Two-Dimensional Compressed GDL

To investigate the transport of liquid water within the GDL, the microscopic pore structure of the GDL needs to be reconstructed. In this paper, the stochastic reconstruction method is adopted to generate a two-dimensional uncompressed GDL. A binary matrix is used to distinguish between solids and pores in the computational domain, with solid points being 1 and pore points being 0. To simplify the GDL structure, the following assumptions need to be made: the fibers are infinitely long and cylindrical with the same diameter and are randomly arranged in the plane, and the orientation with respect to the thickness direction is ignored. The GDL in the calculation domain can therefore be represented by circular particles with the same diameter and without overlapping.

Compression leads to the deformation of the GDL in the region below the ribs. As the compression ratio increases, the pore space of the GDL is compressed, and the deformation of the GDL in the region below the ribs becomes progressively more pronounced, with the ribs intruding into the GDL. Here, the compression ratio (CR) is defined as the ratio of the reduced thickness to the initial thickness [52].
(16)$$CR=\frac{t-t_{0}}{t_{0}}$$

The local compression of the GDL is achieved in this paper by reducing the pore space below the ribs. Deformation of the GDL below the GC can be neglected based on experimental observations and simulation work on cross-sections of compressed GDL by Jeon et al. [53]. Details of the methodology for compression can be found in our previous work [48].

### 2.4. Computational Domain and Boundary Conditions

The proton exchange membrane (PEM), catalyst layer (CL), GDL, and bipolar plate (BPP) are the key components of the PEMFC. As shown in Figure 2a, a representative element in the PEMFC is selected for the computing domain, and the computational domain includes the GDL, the GC, and the rib, with the carbon fibers in the GDL represented by solid circles with the same diameter of 7 lu, here, 1 lu represents 1 µm. The computational domain has a size of 1000 × 400 lu^2^, the height of the gas channel is 194 lu, the thickness of the GDL (y_GDL_) is 196 lu, and there is a 10 lu liquid water entrance area below the GDL. With an average porosity of 0.78, seven GDL structures with different porosity distributions are constructed, corresponding to M1 to M7, as shown in Figure 2b. The porosity of the GDL is uniformly distributed in M1; the porosity of both M2 and M5 (linear) gradually increases, with a larger gradient in M2; M3 and M6 (multilayer) are designed as three layers with increasing porosity along the thickness direction, with a larger gradient in M3; the porosity of both M4 and M7 (inverted V-shaped) increases and then decreases, with a larger gradient in M4. The fibers are divided into 28 layers along the thickness direction, and controlling the number of fibers in each layer allows the design of a porosity gradient along the thickness direction. The contact angle of the carbon fiber is set to 120° [20,25], corresponding to w = 0.013, while the ribs are set to be slightly hydrophobic with a contact angle of 110° [24], corresponding to w = 0.010.

A half-way bounce-back boundary is applied to all solid surfaces to realize the no-slip boundary condition. A periodic boundary is implemented to the left and right planes of the computational domain. The boundary condition proposed by Zou and He [54] is applied to the inlet (bottom of the computational domain) and the outlet (top of the computational domain). The inlet flow velocity of the liquid water is set to 1 × 10^−4^ lu, resulting in a Reynolds number and capillary number of 0.0042 and 0.0025, respectively. The dimensionless number is sufficiently small to indicate that the flow of liquid water within the GDL can neglect viscous effects and that surface tension plays a dominant role in the behavior of liquid water in the GDL. When liquid water within the GDL is capillary-fingered into the flow driven by capillary forces, the viscosity of liquid water and air can be considered equivalent to the SC model. Therefore, in this thesis, the viscosities of liquid, water, and air are the same, and the corresponding relaxation times are 1, that is, τ^air^ = τ^water^ = 1. All programs are established in MATLAB 2022a and run on a server equipped with two 24-core Intel Xeon Gold 6248R (Xiamen, China) 3.0-4.0GHz CPUs and 128GB of RAM in total. 

## 3. Results and Discussion

### 3.1. Effect of Porosity Gradients on the Two-Phase Behavior within Uncompressed GDLs

Designing the GDL pore distribution to improve liquid water distribution within the GDL is becoming one of the emerging methods to improve fuel cell performance. In this section, the two-phase behavior within GDLs with linear, multilayer, and inverted V-shaped porosity gradient distributions is compared with that of porosity uniform GDL. Two gradients are considered for each type of porosity gradient distribution, with local porosities of 0.66 and 0.72 on the catalyst layer side and corresponding local porosities of 0.90 and 0.84 on the GC side, respectively, while ensuring an overall porosity of 0.78. Additionally, we assume that the simulated fuel cell is placed horizontally, which ensures that liquid water forms droplets and liquid films after the breakthrough from the GDL to the gas channel. The PEMFC is considered to operate in isothermal conditions, and evaporation of liquid water is not considered.

Figure 3 shows the liquid water distribution for M1, M2, M3, and M4 at different simulation times. Initially, liquid water penetrates into the GDL from the water entrance area, and some clusters of liquid water form at the bottom of the GDL. At the moment t = 3.0 × 10^5^ lu, the continuous penetration of liquid water causes it to select water clusters with low capillary resistance to form the main flow path. More small clusters of liquid water are formed in M1 than in M2, M3, and M4 due to the greater flow resistance of liquid water caused by the smaller local porosity of M2, M3, and M4 on the CL side. After the formation of the dominant path, other liquid water will prefer this flow path rather than continue to form new clusters from other places of high resistance. It is worth noting that at this time, the liquid water in M3 has broken through the GDL and formed droplets in the GC, which means that the porosity gradient can increase the flow velocity of liquid water. In the third stage, the first dominant pathway in M1 is difficult to break through due to the small local porosity at the end of the pathway. Another dominant pathway is created, and liquid water breaks through to the GC to form droplets. For M2, one of the dominant paths is located below the rib; the liquid water reaches the bottom of the rib and flows to the left and right due to the small capillary resistance caused by the large local porosity, leading to the accumulation of liquid water under the rib. In M3, there are two breakthroughs occurring at the end of the dominant flow paths. One breaks through close to the ribs and forms a water film, another breaks through in the middle of the GC and forms a droplet, and the third water cluster is difficult to circulate due to high capillary resistance. In M4, liquid water saturation (***Sw***) is high in this region due to the large local porosity in the middle of the GDL, which suggests that the porosity gradient distribution has a significant effect on liquid water distribution and transport. The accumulation of liquid water will occupy more of the gas transport path, and there are several liquid water cycles forming here, which will make it difficult for air to be transported to the CL, potentially leading to an inadequate supply of oxygen for electrochemical reactions and reduced fuel cell performance. In the third stage, the breakthrough of liquid water into the GC occurs in M1, M2, and M3, and the distribution of liquid water within the GDLs reaches a steady state. The pressure of liquid water connected within the main flow paths where breakthrough has occurred is significantly less than the capillary forces originating from surface tension. This is because the pressure force from the connecting water within the dominant flow paths is significantly less than the capillary force from surface tension. In the fourth stage, the saturation and distribution of liquid water within the GDL in M1, M2, and M3 no longer changes. Droplets continue to grow in the GC, and sufficient liquid water accumulates in M4 and reaches a steady state. The accumulation of liquid water in M4 is sufficient and reaches a steady state, but the accumulation of liquid water in the middle of the GDL and the formation of the cycle result in a dramatic increase in the time to reach a steady state.

The water transport characteristics of M1, M2, M3, and M4 in uncompressed conditions are shown in Figure 4. The evolution of the total liquid water saturation within the GDL over time is illustrated in Figure 4a. Water saturation values within four samples increase linearly in the earlier period. Breakthrough of liquid water within M1, M2, M3, and M4 occurred at 4.5 × 10^5^ lu, 3.9 × 10^5^ lu, 2.5 × 10^5^ lu, and 8.1 × 10^5^ lu, respectively. The time to the breakthrough of liquid water within M2 and M3 is 34.4% and 44.4% faster than that of M1, respectively, implying that water generated in the catalytic layer can be removed more quickly. The water saturation gradients slow down after the liquid water breakthrough and eventually reach a steady state. When the liquid water distribution within the GDL is stable, M2 and M3 facilitate water removal, with water saturation in M2 and M3 decreasing by 13.9% and 36.9%, respectively, compared to that in M1. In contrast, the time for liquid water to reach a steady state within M4 is dramatically longer, and water saturation increases by 81.3% compared to that within M1, indicating that M4 has a negative impact on water management. Further, the variation of liquid water saturation within the GDL under the channel and under the rib with time is given in Figure 4b,c. The water saturation in the GDL under the channel is significantly higher than that under the rib, indicating that the GDL under the channel is the main region for liquid water transport. The distribution of water saturation along the through-plane direction as liquid water reaches the steady state within M1, M2, M3, and M4 is shown in Figure 4d. According to Laplace’s law, the capillary pressure is negatively correlated with the pore radius. Compared to M1, M2, M3, and M4 have smaller local porosity at the bottom of the GDL, meaning a smaller local pore size distribution, resulting in greater capillary resistance and, therefore, lower local water saturation. For M2 and M3, some of the dominant flow paths of liquid water have formed, and the large capillary resistance in the y/y_GDL_ < 0.2 region prevents more water clusters from forming so that water saturation in the y/y_GDL_ > 0.2 region remains low. The increase in water saturation at the top of M2 is due to the fact that the dominant flow path is below the rib, with liquid water flowing laterally in the top region of the GDL to find a breakthrough point. In contrast, the local porosity in the middle of M4 increases sharply and the local capillary resistance decreases, leading to a pooling of liquid water and blocking the transport path of the reactive gas.

Figure 5 shows the liquid water distribution for M1, M5, M6, and M7 at different simulation times. Compared to M2, M3, and M4, M5, M6, and M7 have a smaller porosity gradient and, therefore, more water clusters can be seen extending from the water entrance area into the GDL in the second stage. In the second stage, the distance from the end of the water clusters to the GC is further within M5 and M6 than within M2 and M3, suggesting that the reduced porosity gradient has also slowed the flow velocity of liquid water within the GDL. Similar liquid water accumulation behavior within the middle of the GDL is also observed within M7.

The evolution of water saturation within M1, M5, M6, and M7 with time is revealed in Figure 6a. The water saturation at a steady state in M5 is not significantly lower than in M1, but the time of liquid water breakthrough is earlier. The time for liquid water to reach a steady state within M6 is 11.1% faster than M1, and water saturation is 13.4% lower, demonstrating the advantages of this structure for water management. The accumulation of liquid water in the middle region of M7 results in a significant increase in both the saturation of liquid water within M7 and the time to reach a steady state. The distribution of water saturation along the through-plane direction as liquid water reaches the steady state within M1, M5, M6, and M7 is shown in Figure 6b. Similarly, in the region where y/y_GDL_ < 0.2, the local water saturation of M5 and M6 is smaller than that of M1, suggesting that smaller local porosity on the catalyst layer side reduces the number of dominant flow paths. The phenomenon can be observed even though the porosity gradient of M5 and M6 is smaller than that of M2 and M3. With the same volume of liquid water flowing in from the entrance area, fewer predominant flow paths mean an increased liquid water flow velocity, which explains the earlier breakthrough of liquid water within M2, M3, M5, and M6 than within M1.

After the liquid water reaches the steady state within the GDLs, the liquid water saturation magnitude is M3 < M6 < M2 < M5 < M1 < M4 < M7. The results show that linear and multilayer porosity gradient distributions can reduce water saturation within the GDL and that the larger the porosity gradient, the lower the water saturation. However, the inverted V-shaped porosity gradient distribution leads to an inappropriate accumulation of liquid water in the GDL, which is detrimental to the removal of liquid water and may block the transport paths of oxygen to the catalyst layer.

### 3.2. Effect of Compression on the Two-Phase Behavior within Porosity Gradient GDLs

In this section, the effect of compression on the liquid water transport processes within the established porosity gradient GDLs is evaluated. According to the assumptions of Jeon et al. [53], compression leads to a reduction in the volume of the pore space while the volume of the fiber is conserved. As the concern is focused on the effect of the GDL being compressed, the deformation of the uncompressed part is not considered. In this study, two different compression ratios (CR = 20%, 40%) are applied to all seven models.

The liquid water distribution at steady state for the seven different porosity gradients of GDL considered at CR = 0, 20%, and 40% is shown in Figure 7. The increase in compression ratio leads to a reduction in porosity and an increase in capillary resistance in the region below the ribs, which prevents the flow of liquid water to the region below the ribs and, therefore, significantly reduces the water saturation below the ribs. For M1, at CR = 20%, the liquid water content in the region below the rib is reduced, while the intrusion of liquid water in the entrance region drives the extension of the liquid water flow path below the channel. At CR = 40%, however, further compression drives the liquid water towards the channel where the capillary resistance is lower. The accumulation of liquid water causes the original capillary resistance to be overcome, leading to the formation of a second breakthrough path, and this process delays the time for the liquid water distribution to reach a steady state. For M2 and M7, the breakthrough path for liquid water at CR = 0 is located below the ribs, and when the GDL is compressed, the initial dominant liquid water flow path is blocked. Liquid water turns to form other dominant flow paths in the region below the channel and eventually breaks through to the GC, suggesting that compression has a crucial influence on the transport of liquid water below the ribs. For M3, M4, M5, and M6, on the other hand, the liquid water breakthrough path is located in the region below the channel when the GDL is not compressed, and there is no significant change in the distribution of liquid water in the region below the channel, regardless of whether it is at CR = 20% or 40%. In the uncompressed samples of M2, M3, M6, and M7, liquid water attaches to the side of the ribs, forming a water film, which is much more difficult to remove during fuel cell operation and will block the oxygen transport pathway to the under-rib region. While in the compressed samples of M2, M3, M6, and M7, the liquid water flow paths are changed due to the compression changing the pore distribution below the ribs, the liquid water flow paths are difficult to form at the rib-GDL interface, and the water film in the GC disappears.

The variation of liquid water saturation with time within the compressed GDL is shown in Figure 8. For GDL with uniform porosity (M1), compression leads to an increase in the saturation of liquid water within the GDL, in line with the results of Jeon et al. [53]. For the seven GDL models studied, it can be seen that the gradient in liquid water saturation increases with increasing compression ratio before the liquid water breaks through to the GC. As the liquid water inlet is set to a constant velocity boundary and the flow rate of liquid water intrusion into the GDL is constant, the increase in the gradient of liquid water saturation may be due to a reduction in pore space due to compression, so the same volume of liquid water will reflect a higher level of water saturation. For M1, M2, M3, M4, M5, M6, and M7, the time for liquid water to break through to GC reduced by 8.9%, 10.3%, 4.0%, 7.4%, 10.5%, 20.0%, and 18.4% at CR = 20%, respectively, compared to the uncompressed samples. Whereas, at CR = 40%, the time for liquid water to break through to GC reduced by 13.3%, 17.9%, 8.0%, 24.7%, 18.4%, 32.5%, and 31.6%, respectively. Compression resulted in a widespread shortening of the time for liquid water to break through into the GC, and the higher the compression ratio, the earlier the breakthrough. This is because the higher the compression ratio, the greater the capillary resistance below the ribs and the more difficult it is for liquid water to penetrate and instead extend below the channel, making the flow velocity of liquid water below the channel faster. The magnitude of liquid water saturation at steady state is M3 < M6 < M2 < M5 < M1 < M4 < M7 for both CR = 20% and 40%, a trend consistent with that of the uncompressed samples, indicating that both linear and multilayer GDLs have superior water management both under compression and uncompressed.

## 4. Conclusions

Water management is a crucial factor limiting the performance improvement of PEMFCs. A two-dimensional pseudo-potential multi-phase LB model is established to study the water transport behavior within the GDL. Seven GDLs with different porosity gradients are generated to investigate the effect of the design of linear, multilayer, and inverted V-shaped porosity distributions on the two-phase behavior within the GDL. Additionally, the effect of compression on the transport of liquid water within the porosity gradient GDLs is investigated. The results show that both porosity gradients and compression have a significant effect on the two-phase behavior within the GDL.

(1)The water saturation within the GDL increases nearly linearly before liquid water breaks through, and the gradient of water saturation slows and eventually reaches a steady state after liquid water breakthrough to GC.(2)The linear and multilayer distribution of porosity in both uncompressed and compressed GDLs leads to a reduction in liquid water saturation, and the larger the porosity gradient, the lower the liquid water saturation. Inverted V-shaped porosity gradient GDLs aggravate the accumulation of liquid water, leading to difficulties in reactive gas transport.(3)Compression led to a reduction in porosity in the under-rib region, resulting in a reduction in the time for liquid water to break through to GC in all samples, suggesting that compression led to faster removal of liquid water.(4)Compression blocks the liquid water transport paths at the rib-GDL interface, indirectly inhibiting the formation of water films on the surface of the ribs.(5)Liquid water saturation is the lowest and liquid water breakthrough to GC is the fastest in multilayered porosity gradient GDLs. Compared to the liquid water saturation and breakthrough time in M1, the reduction is 36.9% and 44.4% for M3 without compression, 34.1% and 41.4% for M3 with CR = 20, and 42.4% and 41.0% for M3 with CR = 40%, respectively.

The porosity gradient GDL has been shown to be beneficial for water management; however, the effects on electron and heat transfer are out of the scope of this paper, which will require a comprehensive model of multi-phase flow, electrochemistry, and heat transfer in future studies. Additionally, extending the model to three dimensions could improve accuracy, but it is currently difficult to achieve due to the dramatic increase in computational resources and will be considered in further work.

## Figures and Tables

**Figure 1 membranes-13-00303-f001:**
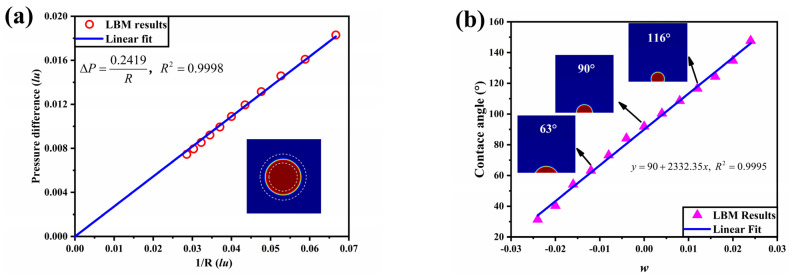
(**a**) The pressure difference as a function of the 1/R; (**b**) The contact angles of the droplet on the solid plane as a function of w.

**Figure 2 membranes-13-00303-f002:**
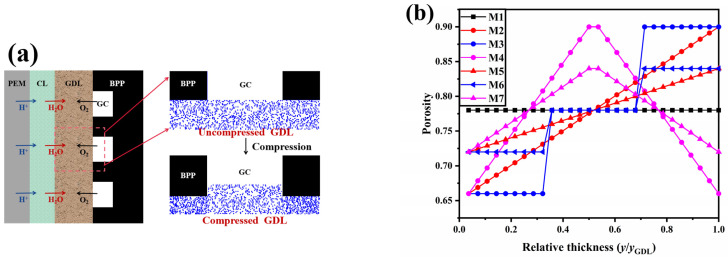
(**a**) Schematic diagram of the computational domain and (**b**) porosity distribution.

**Figure 3 membranes-13-00303-f003:**
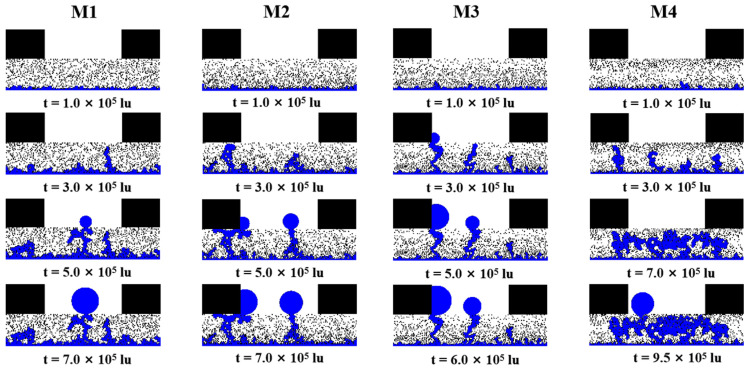
The liquid water distribution for M1, M2, M3, and M4 at different simulation times.

**Figure 4 membranes-13-00303-f004:**
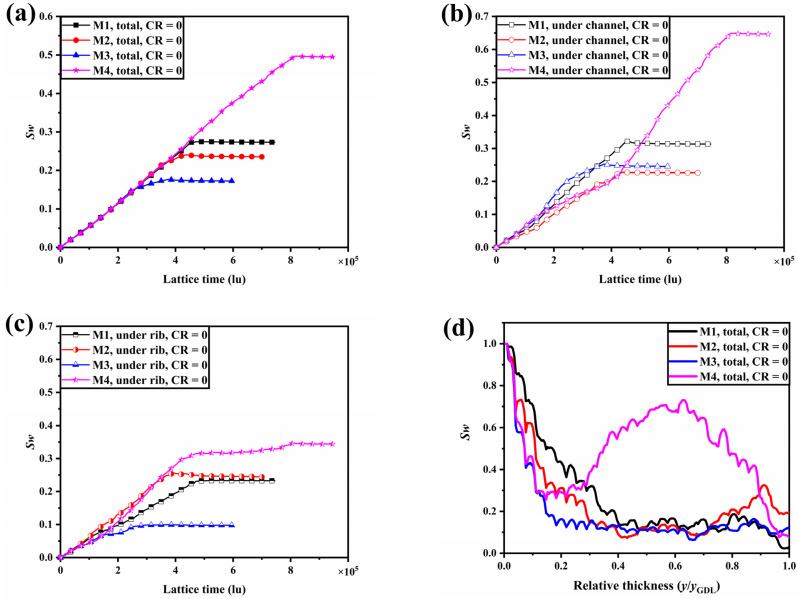
(**a**) The evolution of the total liquid water saturation within the GDL over time; the variation of liquid water saturation within the GDL under the channel (**b**) and under the rib (**c**) with time; (**d**) the distribution of water saturation along the through-plane direction as liquid water reaches the steady state within M1, M2, M3, and M4.

**Figure 5 membranes-13-00303-f005:**
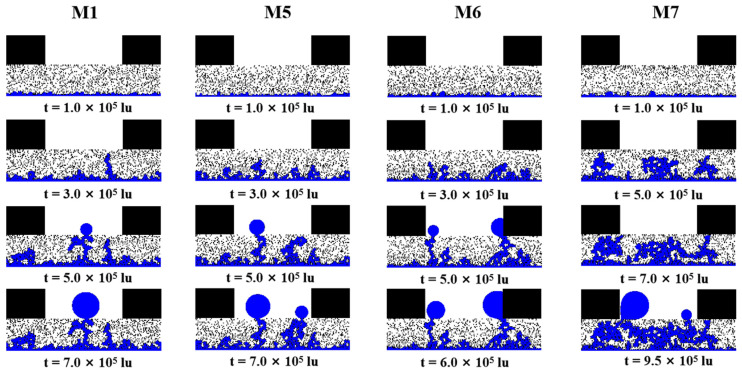
The liquid water distribution for M1, M5, M6, and M7 at different simulation times.

**Figure 6 membranes-13-00303-f006:**
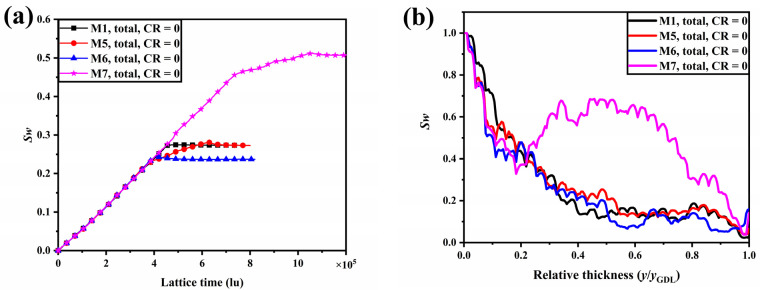
(**a**) The evolution of water saturation within M1, M5, M6, and M7 with time; (**b**) the distribution of water saturation along the through-plane direction as liquid water reaches the steady state within M1, M5, M6, and M7.

**Figure 7 membranes-13-00303-f007:**
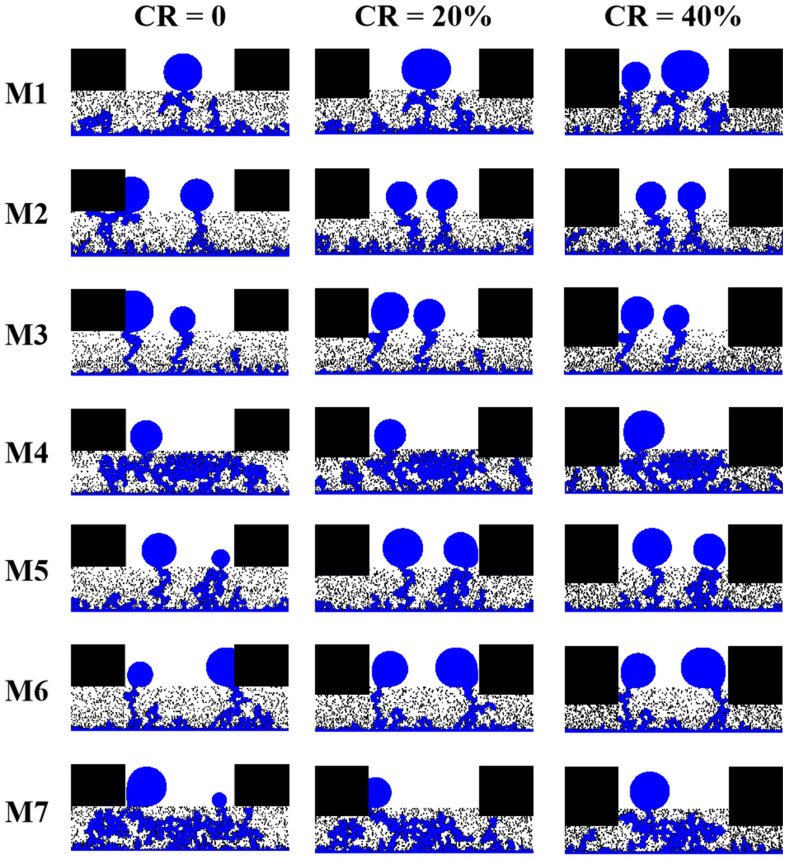
The liquid water distribution at steady state for the seven different porosity gradients of GDL considered at CR = 0, 20%, and 40%.

**Figure 8 membranes-13-00303-f008:**
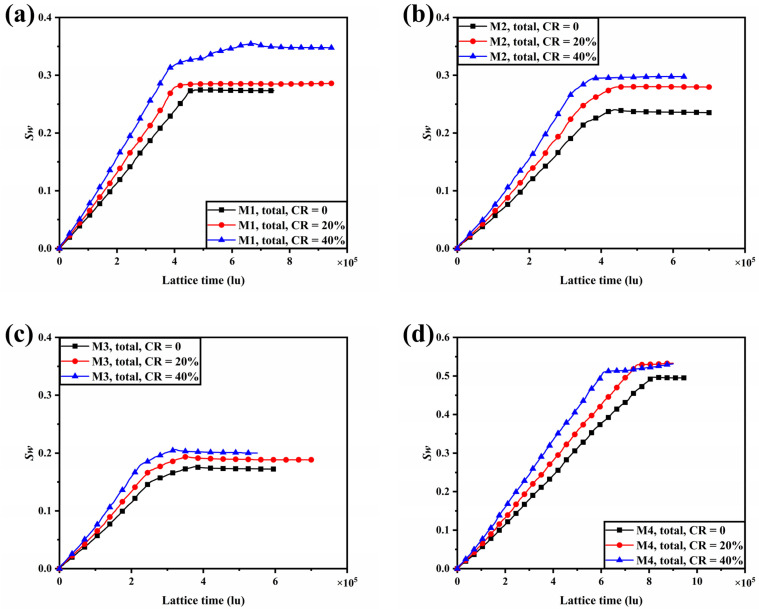
The evolution of water saturation within the seven different porosity gradients of GDL considered at CR = 0, 20%, and 40% with time, (**a**) M1; (**b**) M2; (**c**) M3; (**d**) M4; (**e**) M5; (**f**) M6; and (**g**) M7.

## Data Availability

Not applicable.

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
