# Peer review of "Effects of Compression and Porosity Gradients on Two-Phase Behavior in Gas Diffusion Layer of Proton Exchange Membrane Fuel Cells"

_membranes, 2023, doi:10.3390/membranes13030303_

Round 1

Reviewer 1 Report

In this article water management in gas diffusion layers (GDL) with different structural properties was investigated. The water-air system was simulated using a common lattice Boltzmann method, i.e. the Shan-Chen multi-component model. Water agglomeration and breakthrough are studied and conclusions drawn on how to design GDLs properly.

The article might be of interest to researchers in this field. It is well written, structured, and the figures are chosen appropriately. However, some questions remain. The most severe one is whether it is applicable to reduce a 3D structure to a 2D structure, although the structure has a huge influence on the physical phenomena and results. Another important point is that the model parameters are not given in a way that the results could be reproduced.

All in all, I cannot recommend the publication of this article in its current form.

Author Response

Responses to the reviewer’s comments:

Reviewer 1:

Comment1: From what I understand no model has been “developed”, but the most common type of multi-component Shan-Chen has been used. This should be clearly stated. Otherwise, the authors strongly overstate their work. Having said this, it is unclear why a validation has been carried out, as this  has  been  done  already  in the  literature the  corresponding values  can  be taken from corresponding references.

Response: Thank you for your comments. The descriptions of the LB models mentioned in this paper have been reworded from the manuscript with the word ‘developed’ removed. Model validation is generally considered mandatory in works related to LB models. Due to the lack of commercial software based on LB theory, researchers usually establish program independently to carry out relevant simulations based on equations 1 to 12 in the manuscript. The difference in each person's programming ability and level makes it necessary to demonstrate the feasibility of their code by showing the model verification work they have performed. Also, due to differences in programming habits, some details of the models performed by different researchers may differ. For example, in Section 2.2, we set for the initial state of the liquid water inlet region to be filled with water, set the initial water phase density to 2 and the initial gas phase density to 1×10-5, and set the initial gas phase density to 2 and the initial water phase density to 1×10-5 in the pore space of the GDL and gas channel. The variation of LB parameters is inevitable when the density values are selected differently, but of course there are some other factors that may lead to different parameter selections. Therefore, model validation is necessary.

Comment2:  Eq. (1): The position “x” is a vector and should therefore be plotted bold. Moreover, in Line 164 “cs” should read “cs”; and in Line 166 “ci” should read “ci”.

Response: Thank you for your comments. The errors mentioned have been corrected in the manuscript.

Comment3:  Eq. (9): This equation is not consistent with Eqs. (1)-(8). First, it is unclear what the index “f” means in “Ff”. Moreover, the concept of introducing “x’” is at least questionable. For mathematically being more  correct,  I would  recommend to formulate  everything  by the  index  “i” which gives the different directions. Also “Gkk_ bar (x,x’)” is wrong, because Gkk_bar  can either be evaluated at x or x’ (and not at both positions). Finally, “\rhok  bar_ ” is not evaluated at x but at x’ . The same applies for Eq. (11). Finally, it is not explicitly given how Ff and Fs  related to the F in Eq. (7).

Response: Thank you for your comments. Firstly, equation 9 represents the calculation of fluid-fluid forces while equation 12 (in the latest version of the manuscript) is the calculation of fluid-solid forces. I have replaced them in the manuscript with new symbols Fk f-f and Fk f-s respectively, which are easier to understand, and I have added a description of these two symbols in the manuscript.

The relationship between Fk f-f, Fk f-s and Fk is Fk= Fk f-f +Fk f-s, and is given in the manuscript by equation 8.

The equations you point out for the calculation of Fk f-f, Fk f-s and  need to be given in more specific detail, for example the subscript i represents the direction of the different adjacent grid points, which I think is unnecessary. The equations given in the current manuscript are widely accepted by scholars working on LB-related research and are widely adopted in Ref. [1-4]. They are sufficient for the reader to understand the LB model adopted in this work.

  • Yang M, Jiang Y, Liu J, et al. Lattice Boltzmann method modeling and experimental study on liquid water characteristics in the gas diffusion layer of proton exchange membrane fuel cells[J]. International Journal of Hydrogen Energy, 2022, 47(18): 10366-10380. https://doi.org/10.1016/j.ijhydene.2022.01.115
  • Wang Y, Xu H, Zhang Z, et al. Lattice Boltzmann simulation of a gas diffusion layer with a gradient polytetrafluoroethylene distribution for a proton exchange membrane fuel cell[J]. Applied Energy, 2022, 320: 119248.https://doi.org/10.1016/j.apenergy.2022.119248
  • Guo L, Chen L, Zhang R, et al. Pore-scale simulation of two-phase flow and oxygen reactive transport in gas diffusion layer of proton exchange membrane fuel cells: Effects of nonuniform wettability and porosity[J]. Energy, 2022, 253: 124101.https://doi.org/10.1016/j.energy.2022.124101
  • Zhou C, Guo L, Chen L, et al. Pore-Scale Modeling of Air–Water Two Phase Flow and Oxygen Transport in Gas Diffusion Layer of Proton Exchange Membrane Fuel Cell[J]. Energies, 2021, 14(13): 3812.https://doi.org/10.3390/en14133812

Comment4: Line 211 and Fig. 1b): The parameter controlling the solid-fluid interaction has originally been introduced as capital W in Eqs. (11) and (12). Moreover, typically the correlation between contact angle and W is described by a 3rd-order equation. Why so you use a linear fit?

Response: Thank you for your comments. In fact, w in equation 13 is used to adjust W and ultimately affect the liquid-solid force, just as g in equation 11 is used to adjust  and ultimately affect the liquid-liquid force. For the latter question, in Ref [5], Moslemi et al stated: ‘Fig. 5 demonstrates a liquid droplet in different contact angles on a solid wall, which indicates an almost linear relationship between the adhesion factor and the contact angle.’ In Ref [6], Wang et al stated: ‘A nearly linear relationship between the predicted contact angle and Gks can be observed in Fig. 6.’

  • Moslemi M, Javaherdeh K, Ashorynejad H R. Effect of compression of microporous and gas diffusion layers on liquid water transport of PEMFC with interdigitated flow field by Lattice Boltzmann method[J]. Colloids and Surfaces A: Physicochemical and Engineering Aspects, 2022, 642: 128623. https://doi.org/10.1016/j.colsurfa.2022.128623
  • Wang Y, Xu H, Zhang Z, et al. Lattice Boltzmann simulation of a gas diffusion layer with a gradient polytetrafluoroethylene distribution for a proton exchange membrane fuel cell[J]. Applied Energy, 2022, 320: 119248.https://doi.org/10.1016/j.apenergy.2022.119248

Comment5: Typically, the wetting properties of carbon are more hydrophilic than assumed by the authors. Why have the authors chosen 120° for carbon and 110° for the ribs, respectively? Could the authors please give any reference to the literature, where this has been measured accordingly?

Response: Thank you for your comments.  According to Ref. [7,8], the contact angle of the GDL was measured to be 120°-150°. And in the reported works, Ref. [3,9-13] adopts 120° as the contact angle of the GDL, while Ref. [12, 15-17] adopts 110° as the contact angle of the rib. Corresponding references are marked in the manuscript.

  • Lim C, Wang C Y. Effects of hydrophobic polymer content in GDL on power performance of a PEM fuel cell[J]. Electrochimica Acta, 2004, 49(24): 4149-4156. https://doi.org/10.1016/j.electacta.2004.04.009.
  • Mortazavi M, Tajiri K. Effect of the PTFE content in the gas diffusion layer on water transport in polymer electrolyte fuel cells (PEFCs)[J]. Journal of Power Sources, 2014, 245: 236-244.https://doi.org/10.1016/j.jpowsour.2013.06.138.
  • Javaherdeh K, Moslemi M, Ashorynejad H R. Liquid water phenomena in compressed gas diffusion and micro-porous layers of Proton exchange membrane fuel cell[J]. Heat and Mass Transfer, 2022: 1-20.https://doi.org/10.1007/s00231-022-03267-2.
  • Chen L, Luan H B, He Y L, et al. Numerical investigation of liquid water transport and distribution in porous gas diffusion layer of a proton exchange membrane fuel cell using lattice Boltzmann method[J]. Russian journal of electrochemistry, 2012, 48(7): 712.https://doi.org/10.1134/S1023193512070026.
  • Kim K N, Kang J H, Lee S G, et al. Lattice Boltzmann simulation of liquid water transport in microporous and gas diffusion layers of polymer electrolyte membrane fuel cells[J]. Journal of Power Sources, 2015, 278: 703-717.https://doi.org/10.1016/j.jpowsour.2014.12.044.
  • Fang W Z, Li J, Tao W Q. Two-dimensional pore-scale investigation of liquid water evolution in the cathode of proton exchange membrane fuel cells[J]. Numerical Heat Transfer, Part A: Applications, 2020, 79(4): 261-277.https://doi.org/10.1080/10407782.2020.1845558.
  • Ira Y, Bakhshan Y, Khorshidimalahmadi J. Effect of wettability heterogeneity and compression on liquid water transport in gas diffusion layer coated with microporous layer of PEMFC[J]. International Journal of Hydrogen Energy, 2021, 46(33): 17397-17413.https://doi.org/10.1016/j.ijhydene.2021.02.160.
  • Jeon D H, Kim H. Effect of compression on water transport in gas diffusion layer of polymer electrolyte membrane fuel cell using lattice Boltzmann method[J]. Journal of Power Sources, 2015, 294: 393-405.https://doi.org/10.1016/j.jpowsour.2015.06.080.
  • Nazari M, Shakerinejad E, Kayhani M H. Tailored Surface Wettability of Gas Diffusion Layer in Polymer Electrolyte Membrane Fuel Cells: Proposing a Pore Scale‐Two Phase Design[J]. Fuel Cells, 2018, 18(6): 698-710.https://doi.org/10.1002/fuce.201700097.
  • Shakerinejad E, Kayhani M H, Nazari M, et al. Increasing the performance of gas diffusion layer by insertion of small hydrophilic layer in proton-exchange membrane fuel cells[J]. International Journal of Hydrogen Energy, 2018, 43(4): 2410-2428.https://doi.org/10.1016/j.ijhydene.2017.12.038.
  • Jeon D H. Effect of gas diffusion layer thickness on liquid water transport characteristics in polymer electrolyte membrane fuel cells[J]. Journal of power Sources, 2020, 475: 228578.https://doi.org/10.1016/j.jpowsour.2020.228578.

Comment6: Line 235: The authors state that the Zou He boundary condition was applied. Since, this boundary condition was developed for single-phase flow only, how was it transferred to two-phase flow here?

Response: Thank you for your comments. Zou He boundary condition is widely used in two-phase LB models, as is the case in GDL-related studies [1,5,6,9,11,13-17]. As two distribution functions are adopted in this paper to describe the water and gas phases, it is simply a matter of applying ZouHe boundary conditions to the distribution functions at the inlet and outlet respectively, the relevant equations of which can be found in our previous work [18].

  • Liao J, Yang G, Li S, et al. Study of droplet flow characteristics on a wetting gradient surface in a proton exchange membrane fuel cell channel using lattice Boltzmann method[J]. Journal of Power Sources, 2022, 529: 231245. https://doi.org/10.1016/j.jpowsour.2022.231245

Comment7: Fig. 2: The abbreviations shown in this figure were not properly introduced. Moreover, please indicate the y-direction as well as yGDL  in Fig. 2a). Finally, in the compresses case the GDL seems to enter the GC. Is this what is observed in reality? How do the authors define yGDL  in this case and how do they measure the porosity in cases where there is an overlap between BPP, GC, and GDL?

Response: Thank you for your comments. In section 2.4, yGDL is declared as 196 lu, which is a constant and does not change with compression. In the simulation of the liquid water flow process inside the GDL, compression caused the ribs to intrude into the GDL and significant deformation of the GDL below the ribs occurred. Jeon et al. [14] made experimental observations of cross sections of compressed GDL and assumed in their modelling work that deformation of the GDL below the channel was negligible, as shown in Figure 1, which is the way numerous scholars have dealt with it.

Figure 1 Photos of GDL compression in the experiment and simulation from Ref. [14].

Comment8: How has the compression been modeled?

Response: Thank you for your comments. In the first manuscript submission, content on the compressed model was lacking. In the latest manuscript, we have added section 2.3 'Two-dimensional compressed GDL' to illustrate the reconstruction and compression model of GDL.

Comment9: It is known from the literature and can be seen from this study, that the structure has a huge influence on the physical phenomena and observations. However, a real structure is 3D and not 2D. This should have a large effect also on the water management and the water agglomeration in the middle of the GDL. A 3D structure would create further flow paths and allow the water to flow also  along the third  direction  leading to  less water agglomeration. Therefore,  I would  either recommend to conduct at least one 3D simulation to show that this effect is negligible or to add some literature references showing that 2D simulations are representative and don’t lead to strong artifacts.

Response: Thank you for your comments. Although the increase in computing power has led to significant developments in numerical simulation, it is not currently feasible to extend the computational domain of this work to three dimensions. For example, the computational domain of this thesis includes 400,000 grid points, whereas a 3D computational domain would require 400,000,000 grid points, and given that the 3D LB model has 19 velocity directions, the computational resources required to compute a 3D model are at least 2100 times that of a 2D model, and the available hardware, such as memory, would not be able to support the program.Therefore, to study pore-scale two-phase flow problems, scholars usually have two ways to simplify the computational domain: (1) selecting representative volume elements from within the 3D GDL for simulation, but this method is difficult to include the rib and GC in the computational domain; (2) reducing the 3D computational domain to 2D. We have chosen a simplified 2D model to study the flow of water at complex interfaces such as rib-GDL-GC and the intrusion caused by compression, which we hope you will understand. We have added further references to the 2D model to the manuscript and have already given a vision at the end for the diffusion of the model into 3D.

Comment10: The saturation Sw was not defined properly.

Response: Thank you for your comments. We have added a description of Sw on page 8.

Comment11: The LBM model parameters are not given properly. For example, what relaxation time was chosen? Moreover, please add the conversion factors that allow to transfer the results given in LBM units to SI units.

Response: Thank you for your comments. In the manuscript, we have given that 1lu represents l µm. And in section 2.4, we give the flow rate of liquid water into the GDL as 1, which is consistent with J et al. The corresponding Reynolds number is 0.0042. The consistency of Reynolds numbers in physical and lattice space ensures that the physical quantities are converted and can guarantee that the flow of liquid water within the GDL is driven by capillary forces. When liquid water within the GDL is capillary-fingered into the flow driven by capillary forces, the viscosity of liquid water and air can be considered equivalent for the SC model. Therefore, in this thesis, the viscosities of liquid water and air are the same and the corresponding relaxation times are 1, that is, τairwater = 1. The relevant content has been modified in the manuscript.

Comment12: A     study     that     might     be     interesting     for     the     authors     and     the     reader     (cf. doi:10.1016/j.advwatres.2022.104320)   develops  a   model  to   efficiently  simulate  two-phase transport in the nanoporous binder phase. The model was extensively applied in a recent study (cf. 10.1002/batt.202200090) to simulate two-phase flows in battery electrodes.

Response: Thank you for your comments. We have followed the literature you have provided and there is much of the work that Martin et al have done that we are learning and are tackling. The literature you have provided has been of vital help to us, thanks again!

Thank you again for your comments, which gave us a deeper and more specific understanding of the research, and also corrected the imperfections. Your comments have taken the paper to a higher level, and I sincerely thank you.

Reviewer 2 Report

A manuscript titled "Effects of compression and porosity gradients on two-phase behavior in gas diffusion layer of proton exchange membrane fuel cells" might be considered for publication in the journal Membranes because it is appropriate to the issue. The calculations take place as assumptions of the way of formation and accumulation of water in fuel cells with a polymer-electrolyte membrane. All calculations seem plausible and the presentation and interpretation of the data in the manuscript are convinced.

1. The formation of a film on the surface takes place only when the design is planar and the electrode is placed horizontally, the authors do not take into account the different configurations of PEMFCs. If this is the case, it should be written in the manuscript.

2. Another limitation in starting and operating PEMFCs is the operating temperature, which affects water evaporation or flooding of the electrode with water. The manuscript should clarify whether the case of water evaporation is considered.

Author Response

Reviewer 2:

A manuscript titled "Effects of compression and porosity gradients on two-phase behavior in gas diffusion layer of proton exchange membrane fuel cells" might be considered for publication in the journal Membranes because it is appropriate to the issue. The calculations take place as assumptions of the way of formation and accumulation of water in fuel cells with a polymer-electrolyte membrane. All calculations seem plausible and the presentation and interpretation of the data in the manuscript are convinced.

Comment1:The formation of a film on the surface takes place only when the design is planar and the electrode is placed horizontally, the authors do not take into account the different configurations of PEMFCs. If this is the case, it should be written in the manuscript.

Response: Thank you for your comments. We have stated this assumption in the manuscript.

Comment2:Another limitation in starting and operating PEMFCs is the operating temperature, which affects water evaporation or flooding of the electrode with water. The manuscript should clarify whether the case of water evaporation is considered.

Response: Thank you for your comments. The PEMFC is considered to operate in isothermal conditions and evaporation of liquid water is not considered. We have stated this assumption in the manuscript.

Thank you again for your comments, which gave us a deeper and more specific understanding of the research, and also corrected the imperfections. Your comments have taken the paper to a higher level, and I sincerely thank you.

Round 2

Reviewer 1 Report

The authors have answered most questions in a very satisfactory manner. Thank you for the nice scientific discussion. However, one major question is still open: Why do the authors think 2D simulations are representative for 3D structures here and don’t lead to unphysical results? If this question is also answered and evidence is given, the article can be published.

Author Response

Responses to the reviewer’s comments:

Reviewer 1:

Comment1: The authors have answered most questions in a very satisfactory manner. Thank you for the nice scientific discussion. However, one major question is still open: Why do the authors think 2D simulations are representative for 3D structures here and don’t lead to unphysical results? If this question is also answered and evidence is given, the article can be published.

Response: Thank you for your rigorous scrutiny of this paper, which is crucial to its improvement in quality. I would like to answer you in the following points:

By reducing a three-dimensional model to a two-dimensional one, it is inevitable that changes in the computational domain and changes in pore connectivity will have an effect on the results to a greater or lesser extent. However, it also depends on whether the simplification of the computational domain is acceptable or whether the resulting effects can be neglected. According to our survey, the representation of carbon fibers as round particles is widely adopted and a significant number of scholars consider this simplified form acceptable [1-11]. Further, some authors have calculated the permeability of the computational domain and compared it with empirical equations to verify the usability of the computational domain [7-11].

Table 1 Comparison of GDL permeability and empirical correlations in the references

Ref. 7

Ref. 8

Ref. 9

Ref. 10

Ref. 11

Therefore, we urgently established the code to calculate the permeability of the 2D GDL and present the results as follows. A very small pressure difference is applied above and below the calculation domain to drive the gas flow within the GDL. The flow (top) and pressure (bottom) fields at steady state are shown in the figure and are very close to the results of Ref. [7].

Figure 1. The flow (top) and pressure (bottom) fields at steady state.

The calculated permeability value is also compared with empirical correlations. As can be seen from Table 2, the permeability value of the homogeneous GDL simulated in this paper is close to the predicted value of the K-C correlation and is within an acceptable range. This is another proof of the viability of the generated 2D GDL.

Table 2. Comparison of GDL permeability (×10-12 m2) and empirical correlations

LBM results

K-C correlation

6.1225

6.76

It has to be said that we do take the reviewers' comments very seriously. Since receiving the first round of review comments, we have immediately carried out a study of liquid water transport processes within the 3D GDL. The distribution of liquid water within the 3D GDL is shown in Figure 2. The blue area represents liquid water and the red area represents gas. However, as previously mentioned, the computational costs required to carry out a 3D study are enormous and the time costs are extended. It is difficult for us to discuss in detail the liquid water transport properties within the 3D GDL in this paper, but again this does not detract from the accuracy, innovation and completeness of the work already carried out in the manuscript.

In summary: The 2D computational domain adopted in the manuscript is one that has been widely adopted and accepted by scholars and the authors have illustrated the correctness of the adopted computational domain by comparing the permeability values of the 2D GDL with the predicted values. Liquid water transport within the 3D GDL will be given serious attention, but is difficult to do so in this paper.

Thanks again! We hope that our response will meet your expectations.

We wish you all the best!

Figure 2. The distribution of liquid water within the 3D GDL.

  • Moslemi M, Javaherdeh K, Ashorynejad H R. Effect of compression of microporous and gas diffusion layers on liquid water transport of PEMFC with interdigitated flow field by Lattice Boltzmann method[J]. Colloids and Surfaces A: Physicochemical and Engineering Aspects, 2022, 642: 128623.
  • Wang Y, Xu H, Zhang Z, et al. Lattice Boltzmann simulation of a gas diffusion layer with a gradient polytetrafluoroethylene distribution for a proton exchange membrane fuel cell[J]. Applied Energy, 2022, 320: 119248.
  • Guo L, Chen L, Zhang R, et al. Pore-scale simulation of two-phase flow and oxygen reactive transport in gas diffusion layer of proton exchange membrane fuel cells: Effects of nonuniform wettability and porosity[J]. Energy, 2022, 253: 124101.
  • Javaherdeh K, Moslemi M, Ashorynejad H R. Liquid water phenomena in compressed gas diffusion and micro-porous layers of Proton exchange membrane fuel cell[J]. Heat and Mass Transfer, 2022: 1-20.
  • Jeon D H, Kim H. Effect of compression on water transport in gas diffusion layer of polymer electrolyte membrane fuel cell using lattice Boltzmann method[J]. Journal of Power Sources, 2015, 294: 393-405.
  • Zhou C, Guo L, Chen L, et al. Pore-Scale Modeling of Air–Water Two Phase Flow and Oxygen Transport in Gas Diffusion Layer of Proton Exchange Membrane Fuel Cell[J]. Energies, 2021, 14(13): 3812.
  • Lee S H, Kim H M. Effects of rib structure and compression on liquid water transport in the gas diffusion layer of a polymer electrolyte membrane fuel cell[J]. Journal of Mechanical Science and Technology, 2022: 1-12.
  • Kim K N, Kang J H, Lee S G, et al. Lattice Boltzmann simulation of liquid water transport in microporous and gas diffusion layers of polymer electrolyte membrane fuel cells[J]. Journal of Power Sources, 2015, 278: 703-717.
  • Shakerinejad E, Kayhani M H, Nazari M, et al. Increasing the performance of gas diffusion layer by insertion of small hydrophilic layer in proton-exchange membrane fuel cells[J]. International Journal of Hydrogen Energy, 2018, 43(4): 2410-2428.
  • Jeon D H. The impact of rib structure on the water transport behavior in gas diffusion layer of polymer electrolyte membrane fuel cells[J]. Journal of the Energy Institute, 2019, 92(3): 755-767.
  • Habiballahi M, Hassanzadeh H, Rahnama M, et al. Effect of porosity gradient in cathode gas diffusion layer of polymer electrolyte membrane fuel cells on the liquid water transport using lattice Boltzmann method[J]. Proceedings of the Institution of Mechanical Engineers, Part A: Journal of Power and Energy, 2021, 235(3): 546-562.
